# Effect of Seaweed-Derived Fucoidans from *Undaria pinnatifida* and *Fucus vesiculosus* on Coagulant, Proteolytic, and Phospholipase A_2_ Activities of Snake *Bothrops jararaca*, *B. jararacussu*, and *B. neuwiedi* Venom

**DOI:** 10.3390/toxins16040188

**Published:** 2024-04-12

**Authors:** Camila Castro-Pinheiro, Luiz Carlos Simas Pereira Junior, Eladio Flores Sanchez, Ana Cláudia Rodrigues da Silva, Corinna A. Dwan, Samuel S. Karpiniec, Alan Trevor Critchley, Andre Lopes Fuly

**Affiliations:** 1Department of Molecular and Cellular Biology, Federal Fluminense University, Niterói 24001-970, Rio de Janeiro, Brazil; ccastro@id.uff.br (C.C.-P.); lcarlos@id.uff.br (L.C.S.P.J.); anacrs1@yahoo.com.br (A.C.R.d.S.); 2Laboratory of Biochemistry of Proteins from Animal Venoms, Research and Development Center, Ezequiel Dias Foundation, Belo Horizonte 30510-010, Minas Gerais, Brazil; eladio.flores@funed.mg.gov.br; 3Marinova Pty, Ltd., Cambridge, TAS 7170, Australia; corinna.dwan@marinova.com.au (C.A.D.); sam.karpiniec@marinova.com.au (S.S.K.); 4Independent Researcher, The Evangeline Trail, Highway 1, Paradise, NS B0S 1R0, Canada; alan.critchley2016@gmail.com

**Keywords:** brown seaweed, snake venom, *Bothrops*, fucoidan, *Undaria pinnatifida*, *Fucus vesiculosus*, antivenom

## Abstract

Background: Snakebite envenomation (SBE) causes diverse toxic effects in humans, including disability and death. Current antivenom therapies effectively prevent death but fail to block local tissue damage, leading to an increase in the severity of envenomation; thus, seeking alternative treatments is crucial. Methods: This study analyzed the potential of two fucoidan sulfated polysaccharides extracted from brown seaweeds *Fucus vesiculosus* (FVF) and *Undaria pinnatifida* (UPF) against the fibrinogen or plasma coagulation, proteolytic, and phospholipase A_2_ (PLA_2_) activities of *Bothrops jararaca*, *B. jararacussu*, and *B. neuwiedi* venom. The toxicity of FVF and UPF was assessed by the hemocompatibility test. Results: FVF and UPF did not lyse human red blood cells. FVF and UPF inhibited the proteolytic activity of *Bothrops jararaca*, *B. jararacussu*, and *B. neuwiedi* venom by approximately 25%, 50%, and 75%, respectively, while all venoms led to a 20% inhibition of PLA_2_ activity. UPF and FVF delayed plasma coagulation caused by the venoms of *B. jararaca* and *B. neuwiedi* but did not affect the activity of *B. jararacussu* venom. FVF and UPF blocked the coagulation of fibrinogen induced by all these Bothropic venoms. Conclusion: FVF and UPF may be of importance as adjuvants for SBE caused by species of *Bothrops*, which are the most medically relevant snakebite incidents in South America, especially Brazil.

## 1. Introduction

Snake venom is a complex mixture of organic and inorganic compounds, of which 90–95% are active proteins and peptides that produce diverse toxic effects in humans. Snakebite envenomation (SBE) is a neglected tropical disease (NTD) recognized by the World Health Organization (WHO) since 2017, mainly to allow funds for the development of strategies to reduce mortality or morbidity by 50% by 2030 [1,2]. SBE occurs worldwide with 2.7 million incidents, 138,000 deaths, and 400,000 amputations or deformities annually. It is a serious medical problem mainly affecting impoverished, tropical, and subtropical developing regions and a common occupational injury for farmers and agricultural workers, as well as children [3,4,5]. Epidemiological studies of SBE are scarce, which makes it difficult for governments to develop strategies to improve treatments to prevent deaths or morbidities [1,3]. In Brazil, pit vipers of the genus *Bothrops* are responsible for 87% of the registered incidents, followed by the genus *Crotalus* (rattlesnakes, 7.8%) and *Lachesis* (Bushmasters, 2%). The genus *Bothrops* is the largest among vipers, with around 47 species, and among them, *B. jararaca*, *B. jararacussu*, and *B. neuwiedi* are of medical interest since they cause the highest numbers of severe envenomation incidents [6]. The symptoms of SBE inflicted by these pit vipers share some clinical effects, including pain, inflammation, edema, muscle damage around the affected limb, and drastic systemic disturbances in hemostasis (hemorrhage and coagulopathies), as well as the failure of the kidneys and respiratory and circulatory systems, hypotension, and death [7]. *B. jararaca*, *B. jararacussu*, and *B. neuwiedi* are endemic to South America (mainly Brazil, Paraguay, Bolivia, and Argentina) and are nocturnal and terrestrial [8]. The species of *B. jararaca* and *B. neuwiedi* reach around 120 and 60 cm in length, respectively, while *B. jararacussu* is the largest (160–220 cm). In general, the venoms of *Bothrops* are a mixture of protein/peptide families, such as the snake venom metalloproteases (SVMPs), snake venom serine proteinases (SVSPs), phospholipase A_2_ (PLA_2_) enzymes, L-amino acid oxidases (LAAOs), hyaluronidases, and bradykinin potentiators, which are responsible for the venom toxicity [7,9].

The WHO and Brazilian Ministry of Health, as well as other countries, recommend antivenoms as the only available treatment for SBE [10,11,12,13]. In Brazil, the mono- or polyclonal antivenoms are produced at the Institute Butantan (São Paulo, São Paulo, Brazil), Institute Vital Brazil (Niterói, Rio de Janeiro, Brazil), Immunological Production and Research Center (Piraquara, Paraná, Brazil), and Ezequiel Dias Foundation (Belo Horizonte, Minas Gerais, Brazil) through the hyperimmunization of equines, and the antivenoms are intravenously administered to patients after envenomation. Although antivenoms prevent the death of victims, they have some disadvantages, such as inducing allergic reactions (anaphylaxis) and fever and high costs of production, and they are ineffective in preventing tissue necrosis, thus leading to complications, such as amputation or deformity of the affected limb [13,14]. Furthermore, the delay in administering antivenoms may increase the mortality and morbidity of SBE [14]. Given the complexity of SBE and the limitations of antivenoms, seeking alternative or complementary treatments is of great importance. Plants have been used as herbal medicines to treat SBE since ancient times. Currently, the literature reports many species of plants used in whole or in part with antivenom effects [15,16,17,18]. On the other hand, research testing antivenom molecules derived from marine environments is rare. Seaweed belongs to the Kingdom Protista and is classified into three groups: Chlorophyceae (green algae), Rhodophyceae (red algae), and Phaeophyceae (brown algae). There are approximately 1500 species of brown algae; however, some species of the class Phaeophyceae, such as *Ecklonia*, *Laminaria*, *Undaria*, *Fucus*, *Ascophyllum*, and *Himanthalia*, are perhaps the most studied due to the high content of bioactive compounds, including alkaloids, terpenoids, phytosterols, carotenoids, polyphenols, sterols, proteins, and sulfated polysaccharides [19,20]. These compounds have a wide range of pharmacological properties, including anticoagulant, antioxidant, antiviral, anti-inflammatory, anticancer, antifungal, antibacterial [21,22,23], and antivenom [24,25,26,27] activities, as well as ecological functions [28]. Indeed, seaweed has been used by the pharmaceutical and nutraceutical industries to promote human health or treat diseases [20,29]. Thus, the successful applicability of seaweeds makes them good candidates for drug development to treat a wide range of diseases, including SBE. The literature has described the antagonist effects of polysaccharides from red algae *Laurencia dendroidea* [24], *Palisada flagellifera* [25], and *Chondrus crispus* [26] and green alga *Gayralia oxysperma* [27] against the toxic activities of the pit vipers *Lachesis muta*, *B. jararaca*, and *B. jararacussu*. However, less is known about the inhibitory properties of fucoidans from brown algae. Although data are limited, two studies found that fucoidan from *F. vesiculosus* neutralized necrosis of tissue muscle caused by crude venom and purified myotoxic PLA_2_ from *Bothrops asper* and other crotaline venoms, including *Cerrophidion godmani*, *Atropoides mummifer*, and *Bothriechis schlegelii* [30,31].

Fucoidans are sulfated polysaccharides and have structural backbones primarily composed of sulfated fucose groups. The polysaccharides may also contain galactose, xylose, and arabinose in significant proportions, while other compounds may also be present in extracts, including uronic acid and the monosaccharides glucose, mannose, and rhamnose, in different ratios, depending on the species [20]. The pharmacological properties are reported to be dependent on the molecular weight, chemical structure, sugar composition, and the position of the sulfate group [20].

Building on the current knowledge of fucoidans in SBE, this study assessed high molecular weight fucoidan extracts derived from two different species, *F. vesiculosus* (FVF) and *U. pinnatifida* (UPF), against proteolytic, plasma, or fibrinogen coagulation, and PLA_2_ activities induced by *B. jararaca*, *B. jararacussu*, and *B. neuwiedi* venom.

## 2. Results

### 2.1. Toxicity of FVF and UPF

The in vitro toxicity of FVF and UPF was assayed using the hemocompatibility test. Treating red blood cells (RBCs) with water led to 100% lysis, while treatment with saline produced no lysis. The incubation of FVF and UPF (1500 µg/mL) lysed around 2% of RBC. According to the literature [32], values of hemolysis below 10% mean that the compound is non-toxic, and thus FVF and UPF can be classified as non-hemotoxic molecules.

### 2.2. Inhibitory Effect of FVF and UPF against the Proteolytic Activity of the Venom of B. jararaca, B. jararacussu, and B. neuwiedi

*B. jararaca*, *B. jararacussu*, and *B. neuwiedi* venoms (10–50 µg/mL) hydrolyzed the substrate azocasein in a concentration-dependent manner. One effective concentration (EC) of each venom (30 µg/mL) was incubated for 5 min at 37 °C with saline (positive group) or with 300 µg/mL FVF or UPF. As seen in Figure 1, the proteolytic activity of *B. jararaca* (Figure 1A) and *B. jararacussu* (Figure 1B) venom was diminished by approximately 25% and 50% by FVF and UPF, respectively. FVF or UPF inhibited the proteolytic activity of *B. neuwiedi* venom by 75% (Figure 1C). FVF or UPF alone, in the absence of venom, did not hydrolyze azocasein.

### 2.3. Inhibitory Effect of FVF and UPF on the Coagulant Activity of the Venom of B. jararaca, B. jararacussu, and B. neuwiedi

*B. jararaca*, *B. jararacussu*, and *B. neuwiedi* venom induces plasma and fibrinogen coagulation in a concentration-dependent manner, and the amount of venom able to clot plasma or fibrinogen at 60 s was determined as the minimum coagulating concentration (MCC). One unit of the MCC of *B. jararaca* (30 µg/mL), *B. jararacussu* (60 µg/mL), and *B. neuwiedi* (30 µg/mL) venom was incubated with FVF or UPF for 5 min at 37 °C. An aliquot of each mixture was then added to the plasma. As seen in Figure 2A,C (white columns), *B. jararaca* and *B. neuwiedi* venom (30 µg/mL) mixed with saline (positive groups) clotted plasma at around 60–70 s. However, in the presence of 300 µg/mL FVF or UPF, the coagulation of plasma occurred at 95 s (Figure 2A,C—white columns). In experiments using fibrinogen (Figure 2A–C, black columns), the coagulation caused by 10 µg/mL *B. jararaca* or *B. neuwiedi* venom in the presence of FVF and UPF (100 µg/mL) did not occur until 600 s, which was the maximum period of observation. The fibrinogen coagulation caused by *B. jararacussu* venom (20 µg/mL) was inhibited by 200 µg/mL FVP or UPF (Figure 2B, black columns). The coagulation of plasma caused by *B. jararacussu* venom (60 µg/mL) was not inhibited by FVP or UPF (600 µg/mL). FVF or UPF alone did not clot plasma or fibrinogen.

### 2.4. Inhibitory Effect of FVF and UPF on the PLA_2_ Activity of the Venom of B. jararaca, B. jararacussu, and B. neuwiedi

*B. jararaca*, *B. jararacussu*, and *B. neuwiedi* venom was incubated with saline, and reads of 0.7–0.8 at the absorbance of 740 nm were considered 100% PLA_2_ activity. Each venom (50 µg/mL) was incubated with 25 µg/mL FVF or UPF. The inhibition of PLA_2_ activity after 30 min was around 20%, regardless of the venom tested (Figure 3). There were no statistically significant differences among the groups containing fucoidans. PLA_2_ activity of venoms treated with 20 mM of EDTA vanished, and FVF or UPF in the absence of venom did not have PLA_2_ activity.

## 3. Discussion

This study provides evidence that two fucoidan extracts derived from *F. vesiculosus* and *U. pinnatifida*, namely FVF and UPF, respectively, inhibited proteolytic, plasma, or fibrinogen coagulant and the PLA_2_ activity of *B. jararaca*, *B. jararacussu*, and *B. neuwiedi* venom but with different profiles. These activities are directly involved in local tissue damage and systemic hemotoxicity in SBE. The proteolytic, hemorrhagic, and coagulating activities are caused by the SVMP and SVSP families of enzymes [7,33,34], while edema, inflammation, and muscle damage effects are mainly due to PLA_2_ enzymes [35,36]. Nevertheless, SVMP, SVSP, and PLA_2_ are considered the most important and the major group of active proteins of viper venoms and are thus responsible for producing the most serious symptoms in SBE [34,35]. Therefore, inhibition of the toxic effects due to such a family of enzymes by FVF and UPF is of great importance. In the literature, there are differences in the toxicity of venoms between species of *Bothrops*. The venom of *B. jararaca* is highly hemorrhagic due to the presence of SVMPs, e.g., jararhagin [37] and bothropasin [38]. On the other hand, *B. jararacussu* venom is highly myotoxic due to the activity of two molecules with a PLA_2_ structure: the enzymatically inactive bothropstoxin I (Bthtx-I) and the enzymatically active bothropstoxin II (Bthtx-II) [39,40]. *B. neuwiedi* venom has high coagulant, hemorrhagic, and myotoxic activities [41].

Antivenoms do not neutralize such toxic activities caused by Bothropic venoms [41] or other vipers [42] and may even induce hypersensitive reactions in patients [14]. Thus, molecules able to neutralize the toxic activities of snake venoms are required to improve the recovery of victims of SBE or to prevent sequelae. The SVMP and SVSP families include many enzymes with chemically unique features and intriguing mechanisms of action [33,34]. Therefore, blocking these venom enzymes, which have multiple and complex actions, using a single molecule is a challenge. The sulfated polysaccharide fucoidans derived from seaweed may be a good choice due to their safety and efficacy in humans and attractive physical and chemical properties [43]. Moreover, the ease of development of pharmaceutical forms using fucoidans means that they are good candidates for developing medicines [43,44]. Because conventional antivenoms have limitations [14], Fuly et al. [26] developed a gel containing a galactan polysaccharide from the red alga *Chrondus crispus* that protected mice from hemorrhaging caused by the venom of *B. jararaca* and *B. jararacussu* when the gel was applied topically before or after the injection of venom. The literature also describes the use of an ointment or cream containing fucoidan from the brown algae *F. vesiculosus* and *U. pinnatifida* to treat inflammatory skin diseases, thrombosis, wounds, and burns [45]. These authors also studied the pharmacokinetics of fucoidan after topical application of such formulations and showed that fucoidan penetrated the skin efficiently and had a long half-life in plasma, striated muscle, and skin, with no accumulation in plasma even after the repeated application of fucoidan for five days [45]. SBE is an inflammatory illness, producing serious local toxic effects in humans, and is thus a transdermal formulation using fucoidan is a good strategy to aid commercial antivenoms. Our group has developed a gel using carrageenan from *C. crispus* [26], and the development of this type of formulation using FVF or UPF is a promising pathway for treating the local toxic effects of SBE. Fucoidans of different molecular weights have been investigated to discover how they are metabolized in the processes of absorption, distribution, metabolism, and excretion from marine sponge-, fungal-, and algal-derived compounds [46]. Moreover, fucoidans have high chemical and physical stability, biocompatibility, and biodegradability and show no side effects in humans [47]. On the other hand, a disadvantage of fucoidans is the lack of gelation ability, but this limitation can be overcome by combining fucoidans with polymers, such as chitosan, which may provide them with a positive charge.

The effect of FVF and UPF against the proteolytic, coagulant, and PLA_2_ activities of *B. jararaca*, *B. jararacussu*, and *B. neuwiedi* venom may be due to the presence and quantity of sulfate groups within their structure. They have a negatively charged structure, allowing them to bind to the positively charged active regions of the SVMPs, SVSPs, and PLA_2_ enzymes of *Bothrops* venom. The toxic activities of PLA_2_ enzymes of snake venoms depend on the cofactor Ca^2+^, as well as some specific basic amino acid residues of their active site, such as histidine at position 48 and others located at the C-terminal region of PLA_2_ enzymes [35,36]. On the other hand, SVMPs are the dominant components of these venoms and contain Zn^2+^ in the active site. SVSPs affect the coagulation cascade due to the action of thrombin-like enzymes (TLEs) that have fibrinogenolytic activity, leading to fibrin formation through the cleavage of fibrinogen [34,48]. However, other approaches to inhibit SVSPs have been investigated, such as murine monoclonal antibodies (mAbs) and synthetic peptides based on the sequence of their substrates or natural inhibitors [41]. However, mAbs failed to neutralize the toxins of *B. alternatus* and *B. neuwiedi* venoms [41]. Da Silva et al. [49] designed two synthetic peptides that inhibited SVSPs of *B. jararaca* crude venom and the purified serine protease Batroxobin, but not human trypsin, which is a serine protease. Undoubtedly, such peptides are a promising therapeutic tool for improving the treatment of SBE caused by *Bothrops* species.

The sulfate groups attached to the structure of FVF and UPF are crucial for fucoidans to inhibit the plasma coagulation caused by the venoms of *B. jararaca*, *B. jararacussu*, and *B. neuwiedi*. The negatively charged structure of fucoidans may allow them to bind to the metals Ca^2+^ and Zn^2+^ within the active site of enzymes. Some commercial compounds able to chelate divalent metals, such as EDTA and O-phenanthroline, inhibit the toxic effects of SVMPs. Other reagents, such as p-bromophenacyl bromide (p-BPB) and EDTA, react irreversibly with PLA_2_ enzymes, thus inhibiting the toxic effects dependent on the catalytic activity of PLA_2_, i.e., myotoxicity, hemolysis, and ADP- or collagen-induced platelet aggregation [50,51]. Nonetheless, such commercial inhibitors are considered toxic to humans or, in some cases, aquatic environments. In this manuscript, the PLA_2_ activity of *B. jararaca*, *B. jararacussu*, and *B. neuwiedi* venoms was inhibited by EDTA. Therefore, FVF and UPF may share the inhibitory mechanism with such commercial reagents, without producing toxicity to humans or animals.

The molecular weight of FVF and UPF may be an important feature in the blocking of snake venom enzymes. It is reasonable to postulate that larger chains of polysaccharides would have a higher quantity of sulfate groups and, thus, the ability of FVF and UPF to bind venom enzymes would be higher. Overall, the mechanism of inhibition of compounds FVF and UPF against SVMPs, SVSPs, and PLA_2_ enzymes appears to occur through the binding to divalent metals, and, thus, this information should be considered when developing compounds with antivenom effects. A synthetic inhibitor of PLA_2_ enzymes, Varespladib, was considered in 2019 by the U.S. Food and Drug Administration (FDA) as a drug for SBE [52]. Varespladib is able to bind the hydrophobic channel of enzymatically or non-enzymatically active PLA_2_ enzymes, leading to the inhibition of myotoxicity, hemorrhage, and edema caused by such enzymes [53,54]. Thus, a combination of some compounds with FVF and UPF may be a good strategy to enhance the efficacy of neutralization of symptoms of SBE.

Thus, the chelating action of FVF and UPF fucoidans and the size of their chains are relevant to their antivenom activity. However, the exact mechanism of inhibition by FVF and UPF was not addressed in this work because the experiments were performed using crude venom instead of purified enzymes. Future experiments will explore the mechanism of action of FVF and UPF using purified SVMP, SVSP, and PLA_2_ enzymes.

In this work, 1500 µg/mL FVF or UPF lysed around 2% of RBCs, which is an acceptable value for a candidate molecule [32]. The biotechnology company Marinova provides fucoidans to markets, such as food, medical, and pharmaceutical research, and thus a meticulous quality assurance protocol is performed to ensure the purity and safety for human consumption. FVF and UPF are produced in Marinova’s International Organization for Standardization 9001 (ISO9001) accredited analytical laboratory and chemically characterized using the carbohydrate content and linkage analysis methods that are described by the FDA.

FVF and UPF have been approved by the FDA as being in the “Generally Recognized As Safe” (GRAS) category of food ingredients [55]. Moreover, in some European countries, such as Austria, Belgium, France, Poland, Spain, and the United Kingdom, preparations containing *F. vesiculosus* are also allowed [56]. A fucoidan from the alga *Cladosiphon okamuranus* was given orally to humans and showed no toxicity at doses of 4 g per day for two weeks. Moreover, other administration routes of fucoidan, such as subcutaneous injection, were also assayed and did not have any negative impact on homeostasis [56].

## 4. Conclusions

This manuscript evidenced the inhibitory effect of FVF and UPF against proteolytic, plasma, or fibrinogen clotting and PLA_2_ activities caused by *B. jararaca*, *B. jararacussu*, and *B. neuwiedi* venoms. Despite inhibiting such toxic activities with similar efficacy, FVF and UPF inhibited toxic activities caused by *B. neuwiedi* venom more efficiently. Research using fucoidan has demonstrated potential antivenom activities; however, existing data were limited, warranting further research and, hence, the present study. The utilization of FVF and UPF as an antivenom involves many challenges, such as differences in the growth conditions of algae and pollution of the marine environment, as well as a wide range of species that contribute to the heterogeneity of their properties. Nevertheless, FVF and UPF may contribute to the manufacture and development of new active topical compounds against the toxic effects of venoms and aid commercial antivenoms in treating SBE caused by *Bothrops* species.

## 5. Materials and Methods

### 5.1. Reagents and Venoms

The lyophilized venoms of *B. jararaca*, *B. jararacussu*, and *B. neuwiedi* were provided by the serpentarium of the Ezequiel Dias Foundation (FUNED), Belo Horizonte, Minas Gerais, Brazil. The venom was diluted in saline and stored at −20 °C until it was assayed. Snake venom collection was conducted under the authorization of the Brazilian National System for Genetic Heritage Management and Associated Traditional Knowledge (SISGEN), process number A39CTRI 04E. Human fibrinogen, ethylenediaminetetraacetic acid (EDTA), trichloracetic acid, taurocholic acid sodium salt, and azocasein were purchased from Sigma Chemical Co. (St. Louis, MO, USA). All the other reagents or solvents were research grade.

### 5.2. Toxicity of FVP and UPF

The toxicity of FVP and UPF was assessed by the in vitro hemocompatibility test, as described previously [32]. The fucoidan (1500 μg/mL) and saline solutions were incubated with a 13% (*v*/*v*) suspension of RBC for 3 h at 37 °C. Samples were centrifuged for 3 min at 1800× *g,* and the lysis of RBC was measured by the release of hemoglobin at an absorbance (A) of 578 nm using a microplate reader (VersaMax, Molecular Devices, CA, USA). Hemolysis rates of 100% and 0% were obtained by adding distilled water or saline to RBC, respectively.

### 5.3. Fucoidan Material

*Fucus vesiculosus* fucoidan (FVF, batch No. DPFVF2021001) and *Undaria pinnatifida* fucoidan (UPF; batch No. DPGFS2019537) were provided by Marinova (Cambridge, TAS, Australia). The materials were extracted using a proprietary aqueous extraction process. The fucoidan purity of both samples was >95% (dry weight). The fucoidan component and compositional profile for each material can be found in Table 1. The calculation of fucoidan purity requires several inputs that are determined using a range of spectrophotometric assays. The total carbohydrate content of a hydrolyzed sample was assessed using the phenol–sulfuric technique developed by [57], while the uronic acid concentration was determined in the presence of 3-phenylphenol, based on the method by [58]. Sulfate content was analyzed using a BaSO_4_ precipitation method [59]. UPF and FVF were dissolved in saline solution to perform the assays.

### 5.4. Effect of FVF and UPF on the Proteolytic Activity of B. jararaca, B. jararacussu, and B. neuwiedi Venom

The proteolytic activity of *B. jararaca*, *B. jararacussu*, and *B. neuwiedi* venom was performed according to [60]. Different concentrations of each venom (10–50 μg/mL) were incubated with 400 µL azocasein (0.2% *w*/*v* dissolved in 20 mM Tris-HCl, 8 mM CaCl_2_, pH 8.8) for 90 min at 37 °C in a final volume of 800 µL. After 90 min, 400 µL of trichloracetic acid (10%) was added to the medium to stop the reaction. The tubes were centrifuged at 12,000 rpm for 3 min. A total of 1.0 mL of the supernatant was removed and transferred to tubes containing 0.5 mL NaOH 2N. The tubes were read in a spectrophotometer at 420 nm. The amount of venom (µg/mL) able to produce reads of 0.2 (i.e., 70–80% of the maximal activity) was determined as an arbitrary unit, i.e., the effective concentration (EC). One EC of each venom was incubated with FVF or UPF or saline (positive groups) for 5 min at 37 °C at a 1:10 ratio (*w*/*w*) of venom to polysaccharide. After incubation, an aliquot was removed and added to the reaction, and the proteolytic activity was measured, as described above. Negative controls were performed by adding saline or fucoidan to the medium in the absence of venom.

### 5.5. Effect of FVF and UPF on the Coagulant Activity of B. jararaca, B. jararacussu, and B. neuwiedi Venom

The coagulant activity of *B. jararaca*, *B. jararacussu*, and *B. neuwiedi* venom was determined using a digital Amelung coagulometer, model KC4A (Labcon, Heppenheim, Germany). A total of 200 μL of a pool of citrated human plasma from the blood bank of the Hospital Universitário Antônio Pedro of the Federal Fluminense University (HUAP-UFF), under the authorization of the Committee for Ethical in Experimentation (CEP-UFF, CAAE: 28941314.0.0000.5243), or commercial fibrinogen (2 mg/mL) were kept for 1 min at 37 °C Then, different concentrations of *B. jararaca*, *B. jararacussu*, and *B. neuwiedi* venom (2–100 μg/mL) were added to plasma or fibrinogen, and the coagulation time was monitored in seconds in the coagulometer. The amount of venom (μg/mL) able to clot plasma or fibrinogen at 60 s was determined as an arbitrary unit, i.e., the minimum coagulant concentration (MCC). Then, one MCC of each venom was incubated for 5 min at 37 °C with saline (positive group) or UPF or FVF at a 1:10 ratio of venom to polysaccharide (*w*/*w*). An aliquot of each mixture was added to plasma or fibrinogen, and coagulation was monitored as described above. The negative control groups contained solely saline or FVF or UPF in the reaction in the absence of any venom.

### 5.6. Effect of FVF and UPF on the Phospholipase A_2_ Activity of B. jararaca, B. jararacussu, and B. neuwiedi Venom

The PLA_2_ activity of *B. jararaca*, *B. jararacussu*, and *B. neuwiedi* venom was measured as in [61]. Fresh egg yolk obtained from a local supermarket was filtered and diluted in saline to a final volume of 100 mL. One volume of this solution was diluted in nine parts saline. The medium reaction contained 580 µL saline, 25 µL sodium taurocholate 0.4%, 25 µL Tris-HCl (200 mM), pH 7.5, 20 µL CaCl_2_ (0.5 M), and 50 µL of each venom in the presence or absence of FVF or UPF. The enzymatic reaction was initiated by adding 300 µL of the egg yolk substrate solution, and the tubes were read at 740 nm after 30 min. A total of 100% PLA_2_ activity of venoms was achieved by incubating venom with saline, in the absence of FVF or UPF, while 0% PLA_2_ activity was determined in a solution containing solely saline or FVF or UPF in the absence of venom. A total of 20 mM of EDTA (final concentration) was added to the reaction in the presence of *B. jararaca*, *B. jararacussu*, and *B. neuwiedi* venom and in the absence of FVF or UPF, followed by the PLA_2_ activity.

### 5.7. Statistical Analyses

Results are expressed as means ± standard error of the mean (SEM) obtained with the number of experiments indicated in each result using the GraphPad Prism^®^ program. The statistical significance of differences among experimental groups was evaluated using Student’s *t*-test or the Mann–Whitney test. *p*-values < 0.05 were considered statistically significant.

## Figures and Tables

**Figure 1 toxins-16-00188-f001:**
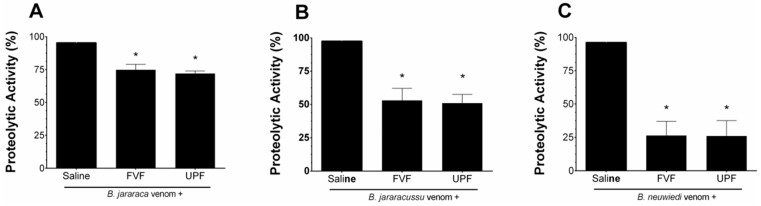
Inhibitory effect of FVF or UPF on the proteolytic activity of *B. jararaca*, *B. jararacussu*, and *B. neuwiedi* venom. A total of 30 µg/mL of *B. jararaca* (**A**), *B. jararacussu* (**B**), and *B. neuwiedi* (**C**) venoms were incubated with saline or 300 µg/mL FVF or UPF for 5 min at 37 °C. An aliquot of each mixture was then added to the reaction, and the proteolytic activity was determined. A total of 100% proteolytic activity was obtained with venom plus saline. Results are expressed as means ± SEM (*n* = 6). * *p* < 0.05 when compared to each venom plus saline.

**Figure 2 toxins-16-00188-f002:**
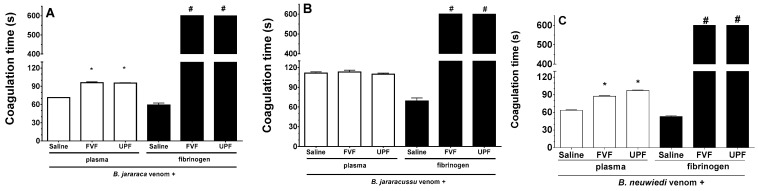
Inhibitory effect of FVF and UPF on the coagulant activity of *B. jararaca*, *B. jararacussu*, and *B. neuwiedi* venom. *B. jararaca* (**A**), *B. jararacussu* (**B**), and *B. neuwiedi* (**C**) venom was incubated for 5 min at 37 °C with saline or with FVF or UPF at a 1:10 ratio of venom to polysaccharide. After incubation, an aliquot was added to plasma (white columns) or fibrinogen (black columns), and the coagulation time was monitored in seconds (s). Results are expressed as means ± SEM (*n* = 8). # means that plasma or fibrinogen did not clot by 600 s of monitoring. * *p* < 0.05 when compared to each venom plus saline.

**Figure 3 toxins-16-00188-f003:**
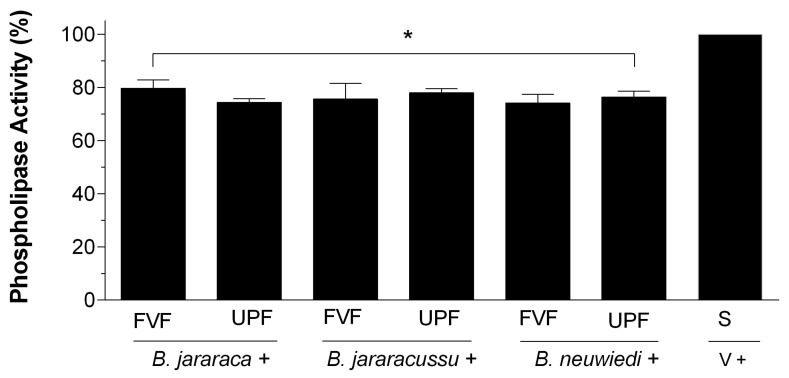
PLA_2_ activity of *B. jararaca*, *B. jararacussu*, and *B. neuwiedi* venom in the presence of FVF or UPF. A total of 50 µg/mL *B. jararaca*, *B. jararacussu*, or *B. neuwiedi* venom was incubated with 25 µg/mL FVF or UPF for 5 min at 37 °C. An aliquot of each mixture was then added to the reaction, and PLA_2_ activity was measured. A total of 100% PLA_2_ activity was achieved by incubating each venom (V) with saline (S) in the absence of FVF or UPF. Results are expressed as mean ± SEM of *n* = 6, considering * *p* < 0.05 when compared to the positive group (V + S).

**Table 1 toxins-16-00188-t001:** Absolute mass percentages of the components of FVF and UPF.

Components	FVF	UPF
Fucoidan	96.1	97
Total carbohydrates	71.7	60.5
Fucose	49.5	28.2
Galactose	2.7	25.6
Uronic acid	3.8	0.8
Polyphenol	<2	<2
Sulfate	30.7	30.4
Cations	~5	6.8

## Data Availability

Data are contained within the article.

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
