# Peer review of "Effect of Seaweed-Derived Fucoidans from Undaria pinnatifida and Fucus vesiculosus on Coagulant, Proteolytic, and Phospholipase A2 Activities of Snake Bothrops jararaca, B. jararacussu, and B. neuwiedi Venom"

_toxins, 2024, doi:10.3390/toxins16040188_

Round 1

Reviewer 1 Report

Comments and Suggestions for Authors

The authors focus their study on the possibility of using fucoidan from Undaria pinnatifida and Fucus vesiculosus to combat the effect of snake venom. It is interesting to have new alternatives at our disposal.

In the text it talks about 600 sg (line 146 and figure), wouldn't it be simpler to express the units in hours?

I do not understand why the possibility of using FVF and UPF in food appears at the end of the discussion, because the title of the article refers to their activity against the action of snake venom.

Reference is made to calcium-mediated activity, why in the experimental development the authors did not analyse the activity of phospholipase A2 under varying calcium conditions? Or against chelating solutions? Since it is alluded to in the discussion?

The authors should elaborate on their results, although it is good that they rely on other authors to confirm their conclusions.

In the conclusions section, the difference in FVF and UPF activity is not mentioned.

Author Response

Reviewer 1

Comments and Suggestions for Authors

The authors focus their study on the possibility of using fucoidan from Undaria pinnatifida and Fucus vesiculosus to combat the effect of snake venom. It is interesting to have new alternatives at our disposal.

1) In the text it talks about 600 sg (line 146 and figure), wouldn't it be simpler to express the units in hours?

Answer: I thank you for your comment. However, the International System of Units (SI) of time is second (s). Thus, we decide to use second. Moreover, the coagulometer (Amelung KC4A) records the coagulation time in seconds.

2) I do not understand why the possibility of using FVF and UPF in food appears at the end of the discussion, because the title of the article refers to their activity against the action of snake venom.

Answer: I thank you for your comment. The sentence explaining that the Industry Marinova provides FVF and UPF in food at the end of the discussion was just to highlight the non-toxicity, the high grade of purity, and ease of cultivation of FVF and UPF. At my point of view, these issues are important due to facilitate the development of new antivenom therapeutic, which is the main topic of this manuscript. On the other hand, this sentence does not mean that eating FVF and UPF will provide protection against the toxic effects of snake bite envenoming.

3) Reference is made to calcium-mediated activity, why in the experimental development the authors did not analyse the activity of phospholipase A2 under varying calcium conditions? Or against chelating solutions? Since it is alluded to in the discussion? The authors should elaborate on their results, although it is good that they rely on other authors to confirm their conclusions.

Answer: I appreciate your comment. We have performed experiments to address this issue about using chelating agent(s) in the PLA2 activity of venoms. We added EDTA (20 mM, final concentration) in the medium reaction, and then, the PLA2 activity of venoms was performed. A graphic was constructed and result is shown below. EDTA reduced around 80-95% the PLA2 activity of B. jararacussu, B. neuwiedi, and B. jararaca venoms. Thus, as already mentioned in the literature by other authors, EDTA is a chelating compound, able to inhibit the enzymatic activity of PLA2 of venoms. As seen in the figure, EDTA inhibited PLA2 activity of B. jararaca, B. jararacussu, and B. neuwiedi venoms.

            I did not put this graphic in the manuscript; however the sentence “In this manuscript, PLA2 activity of B. jararaca, B. jararacussu, and B. neuwiedi venoms was inhibited by EDTA. “ was included at discussion section, pg 7.

Phospholipase A2 activity of B. jararacussu, B. neuwiedi, and B. jararaca venoms in the presence of saline (white columns) or 20 mM EDTA (black columns). Results are expressed as mean ± SEM of n = 6, considering * P < 0.05, when compared to venoms in presence of saline (white columns).

4) In the conclusions section, the difference in FVF and UPF activity is not mentioned.

Answer: I thank you for the comment and I agree with the reviewer. The sentence “The manuscript evidenced the inhibitory effect of FVF and UPF against proteolytic, plasma or fibrinogen clotting, and phospholipase A2 activities caused by B. jararaca, B. jararacussu, and B. neuwiedi venoms. Despite inhibiting such toxic activities with similar efficacy, it seems that FVF and UPF inhibit more efficiently toxic activities caused by B. neuwiedi venom.” was included in the conclusion section to explore better the difference between FVF and UPF activity.

Reviewer 2 Report

Comments and Suggestions for Authors

I think it is great to see different approaches being attempted and the authors have looked at important facets of venom activity to assay for effect. 

One question I think needs to be answered before accepting the paper is what was the purpose of DMSO? DMSO can neutralize venom components by itself and it is necessary to know what it was used for and how (it is only mentioned once). 

Otherwise, I don't have major concerns. If the authors are contemplating this as a topical composition, then I think it is to their benefit to mention this as early as possible as this is an under-represented approach for new therapeutics. It is hard to tell how much the authors believe this is a possibility. 

Nice work. 

Comments on the Quality of English Language

Minor editing. Overall it is fine. 

Author Response

Reviewer 2

Comments and Suggestions for Authors

I think it is great to see different approaches being attempted and the authors have looked at important facets of venom activity to assay for effect.

1) One question I think needs to be answered before accepting the paper is what was the purpose of DMSO? DMSO can neutralize venom components by itself and it is necessary to know what it was used for and how (it is only mentioned once).

Answer: I thank you for your comment and I do agree with the reviewer. Undoubtedly, DMSO inhibits the activity of enzymes of venoms. However, in some cases, low concentration of DMSO is required to dissolve nonpolar compounds, in which was not the situation of this work, since FVF and UPF are polar molecules; and, therefore, they are easily dissolved in saline or water. In this manuscript, we dissolve FVF and UPF in physiological saline. Thus, in this manuscript, we did not use DMSO to dissolve FVF and UPF or in any other part of experiments. The inclusion of DMSO in the section 5.1 (reagents and venoms) was made wrongly. It was my fault. I apologize for the mistake, and DMSO was removed from the manuscript.

2) Otherwise, I don't have major concerns. If the authors are contemplating this as a topical composition, then I think it is to their benefit to mention this as early as possible as this is an under-represented approach for new therapeutics. It is hard to tell how much the authors believe this is a possibility.

Answer: Literature has reported that antivenoms efficiently block the toxic systemic effects, including hemorrhage, coagulation disturbs, neurotoxic, as well as death. The cases of envenomings worldwide are extremely high (around 1 million of cases), but not deaths; perhaps 90,000, with a mortality rate of 0.4 to 1.1%. Likewise, antivenom serotherapy is of great importance to mankind. However, such treatment has some disadvantages, as inefficacy to prevent tissue damage, morbidities, and amputation of the affected limb. Therefore, new approaches should be developed to counteract such local toxic effects, and a topical compound seems to be a good strategy to avoid or reduce such physical disabilities caused by envenomings. It is well know that amputations or any physical disability have a high incidence worldwide, leading to health, psychological, and economics long-term problems to nations. Thus, it is a challenge to deal with the complications of envenomings.

Our group has described antivenom effect of a polysaccharide from the red seaweed Chondrus crispus. A gel with this compound was prepared, and topically applied into mice, resulting in inhibition of hemorrhaging caused by B. jararaca and B. jararacussu venoms (da Silva et al., 2020, reference 26). The efficient neutralization of hemorrhage by this gel opens a gap for the development of other algae-based-gels, for example using FVF and UPF. Therefore, seeking topical compounds able to neutralize local effects of snakebite envenoming, including tissue damage is of great importance; and we do believe that development of a gel with FVF and UPF is a real possibility, due to our previous result (reference 26). This possibility of using FVF and UPF as topical compounds is already described in the discussion, pg 6. The discussion of efficacy and disadvantages of antivenoms, as well as the inhibitory properties of the gel of C. crispus are already described in the manuscript, as well.

Comments on the Quality of English Language

Minor editing. Overall it is fine.

Round 2

Reviewer 1 Report

Comments and Suggestions for Authors

The corrections introduced seem appropriate to the comments previously made. Only one remark about the time units in the SI with the authors. The use of the International System of Units (SI) of time is second (s) seems correct to me. Then the abbreviation “sec” should be modified to “s”. And, does it only refer to seconds to highlight the results and express the experimental conditions with the unit of minutes?

And I think the article can be published

Author Response

Reviewer 1 (Round 2)

The corrections introduced seem appropriate to the comments previously made. Only one remark about the time units in the SI with the authors. The use of the International System of Units (SI) of time is second (s) seems correct to me. Then the abbreviation “sec” should be modified to “s”. And, does it only refer to seconds to highlight the results and express the experimental conditions with the unit of minutes?

And I think the article can be published

Answer: I thank you for your comment and suggestion. The abbreviation “sec” was changed to “s”. The modification in the manuscript was performed, accordingly.

The experiment of coagulation was performed in a digital coagulometer; and the results are registered on its screen in second(s). Thus, we decided to express the results (section 2.3 and Figure 2) of coagulation of plasma and fibrinogen in second(s). The description of experimental condition of coagulation (section 5.5) is also referred as “s”. On the other hand, the description of incubation of venom with saline or with FVF and UPF was expressed in minutes (sections 2.3 and 5.5).

The manuscript was proofread by the Company Cambridge Proofreading & Editing LCC. As suggested by this office, the title of the manuscript was changed to: Effect of seaweed-derived fucoidans from Undaria pinnatifida and Fucus vesiculosus on coagulant, proteolytic, and phospholipase A2 activities of snake Bothrops jararaca, B. jararacussu, and B. neuwiedi venom. We decided to accept the suggestion of this modification.
